# Genetic Diversity Evaluation and Conservation of Topmouth Culter (*Culter alburnus*) Germplasm in Five River Basins in China

**DOI:** 10.3390/biology12010012

**Published:** 2022-12-21

**Authors:** Miao He, Di-An Fang, Yong-jin Chen, Hai-bo Sun, Hui Luo, Ya-fei Ren, Tian-you Li

**Affiliations:** 1National Demonstration Center for Experimental Fisheries Science Education, Shanghai Ocean University, Shanghai 201306, China; 2Key Laboratory of Freshwater Fisheries and Germplasm Resources Utilization, Ministry of Agriculture and Rural Affairs, Freshwater Fisheries Research Center, Chinese Academy of Fishery Sciences, Wuxi 214081, China; 3Wuxi Fisheries College, Nanjing Agricultural University, Wuxi 214081, China

**Keywords:** *Culter alburnus*, germplasm resources, molecular markers, genetic characteristics

## Abstract

**Simple Summary:**

To better protect and manage the germplasm resources of the topmouth culter (*Culter alburnus*), it is important to explore the current status of its genetic diversity. In this study, the mitochondrial DNA *COI* gene was used to analyze the genetic diversity, genetic differentiation, and historical dynamics of *C. alburnus* from five river basins in north and south China. The results showed that the genetic diversity is polarized in different populations but without new species differentiation and has significant geographical features, which are important for developing conservation strategies.

**Abstract:**

To study the genetic diversity of *Culter alburnus* (*C. alburnus*) populations, we analyzed the genetic diversity of five *C. alburnus* populations from Songhua Lake (SH), Huaihe River (HH), Changjiang River (CJ), Taihu Lake (TH), and Gehu Lake (GH) based on mitochondrial *COI* gene sequences. The results showed that the average contents of bases T, C, A, and G in the 526 bp *COI* gene sequence were 25.3%, 18.1%, 28.1%, and 28.6%, respectively, which showed AT bias. A total of 115 polymorphic sites were detected in the five populations, and 11 haplotypes (Hap) were defined. The nucleotide diversity (*P_i_*) of the five populations ranged from 0.00053 to 0.01834, and the haplotype diversity (*H_d_*_)_ ranged from 0.280 to 0.746, with the highest genetic diversity in the TH population, followed by the SH population, with lower genetic diversity in the HH, CJ and GH populations. The analysis of the fixation index (Fst) and the genetic distance between populations showed that there was significant genetic differentiation between the SH population and the other populations, and the genetic distances between all of them were far; the genetic diversity within populations was higher than that between populations. Neutral tests, mismatch distributions, and Bayesian skyline plot (BSP) analyses showed that the *C. alburnus* populations have not experienced population expansion and are relatively stable in historical dynamics.

## 1. Introduction

*Culter alburnus* belongs to the genus Culter within the family Cyprinidae of the order Cypriniformes, which is a large fish in the Culterinae subfamily that is widely distributed in major freshwater lakes and reservoirs. It has a high economic value and ecological significance; it maintains the stability of the water ecosystem and is an important freshwater economic fish in China [1]. However, with continuing human fishing and habitat destruction, *C. alburnus* is facing decreases in natural resources and population genetic diversity, reducing germplasm resources [2].

In recent years, studies on *C. alburnus* were focused on its nutritional composition and feeding habits, anatomical and biological characteristics, anthropogenic reproduction and embryonic development, and genetic diversity [3,4]. With the development of DNA sequencing technology, mitochondrial molecular markers have been widely used for population genetic structure and phylogenetic development analyses [5,6]. Mitochondrial DNA is maternally inherited and is not easy to recombine. It evolves relatively quickly, so it is the preferred molecular genetic marker for population differentiation [7]. *COI* (cytochrome subunit I) is mainly used in species identification and population genetic analyses with close genetic relationships, where it is a commonly used genetic marker method in populations [8]. In previous studies, the genetic diversity of wild *C. alburnus* populations in the Yangtze River was found to be low, closely related, and highly differentiated, which is probably due to a combination of environmental and human effects [9]. There are great genetic differences between Danjiang River and Xingkai Lake. The genetic diversity of Xingkai Lake is low, which may be related to the geographical isolation and habitat differences [10]. The overall genetic diversity of *C. alburnus* in the Pearl River basin is high, but there are still many sampling sites where genetic diversity is at a low level [11]. Therefore, the study of *C. alburnus* genetic diversity and the management of *C. alburnus* germplasm conservation has become a high-priority research area. However, most of the studies on wild *C. alburnus* have focused on only local water bodies, and the sample sizes and geographic diversity of populations have been limited. Few studies have been reported on large geographic scales.

In this study, the genetic diversity and genetic characteristics of five different populations of *C. alburnus* in the Yangtze River, Taihu lake, Gehu lake, Huaihe lake, and Songhua lakes were investigated based on the mitochondrial DNA *COI* gene sequences. The genetic differences among the populations were revealed to provide basic data for the conservation and management of *C. alburnus* germplasm resources.

## 2. Materials and Methods

### 2.1. Experimental Materials

Five *C. alburnus* populations were collected in 2021 and 2022 using gill nets:32 from the Changjiang River (CJ), 31 from the Taihu Lake (TH), 31 from Gehu Lake (GH), 30 from the Huaihe River (HH), and 44 from the Songhua Lake (SH) (Figure 1 and Table 1). All individual fish fins were stored in 95% ethanol until DNA extraction. All experimental procedures were performed in accordance with the regulations for the administration of laboratory animal affairs. Fish sampling was approved and cleared by the China Fishery Resources Commission.

### 2.2. DNA Extraction, PCR Amplification, and Sequencing

About 30 mg tissue samples of *C. alburnus* fins were cut in centrifuge tubes, and DNA was extracted using a kit method (Marine Animal Tissue Genomic DNA Extraction Kit, Tiangen Biochemical Co., Ltd., Beijing, China). The concentration and purity of the DNA were detected by agarose gel electrophoresis at a concentration of 1%, and DNA samples that met the requirements for subsequent amplification were stored at −20 °C for backup [12].

The primer sequences used for the mitochondrial *COI* gene amplification in *C. alburnus* [13] were L5956-*COI*: 5′-CACAAAGACATTGGCACCCT-3′ and H6558-*COI*: 5′-CCTCCTGCAGGGTCAAAGAA-3′as primers in a 50 μL reaction volume mixture containing 25 μL of Premix Taq (1.25 U of rTaq polymerase, a dNTP mixture (each at 0.4 mM), and 4 mM Mg2+; TaKaRa, Dalian, China), 1.0 μL of each of 10 μM primer, 3.0 μL of total genomic DNA (approximately 100 ng as a template), and 18–22 μL of sterile distilled water. PCR was performed according to the reaction method as follows: pre-deformation at 94 °C for 5 min; denaturation at 94 °C for 35 s, annealing at 50 °C for 35 s, and extension at 72 °C for 50 s for 35 cycles; extension at 72 °C for 10 min; end at 4 °C. Purified PCR products were directly sequenced using primers from both ends in a semi-automated DNA analyzer (3700; Applied Biosystems, Waltham, MA, USA).

### 2.3. Genetic Diversity Analysis

After sequencing was completed, the COI gene sequences were compared and corrected using Clustal W software, and the base content composition of the sequences was analyzed using MEGA11 software to calculate the genetic distances within and between populations [14]. DnaSP v5.1 software was used to analyze the genetic diversity index [15]. The Fst (fixation index) was calculated using Arlequin 3.5 software. [16,17]. MEGA11 software was used to construct an ML(maximum-likelihood) tree based on the haplotypes and an NJ (neighbor-joining) tree based on genetic distance. The Kimura two-parameter method was used as the model, and the bootstrap analysis was tested 1000 times. Culter dabryi was selected as the out group, which was conducive to the comparative analysis of the relationships between populations [18]. Phylograms between haplotypes were constructed using Network software based on the median joining equation method [19]. DnaSP v5.1 software for the neutral detection and mismatch distribution analysis of historical population dynamics used the formula t = Tτ/2u (T, generation time; τ, expansion time parameter; u, mutation rate of the sequence) [20]. Beast1.7.3 used Bayesian skyline plots (BSP) to detect population history and effective population size changes. The GTR model was chosen, and sequences were run at a 1% mutation rate per million years, operated for 2 × 10^8^ generations, and merged with Log Combiner. Tracer 1.5 software was used to detect the results of an effective sample size greater than 200 (ESS > 200), which were output as Bayesian skyline plots [21].

## 3. Results

### 3.1. Genetic Diversity

After the *COI* gene sequences were proofread and edited by Clustal W, 168 gene sequences with 526 bp lengths were obtained, with 399 invariant sites, 46 parsimony-informative sites, and 115 variant sites. The average contents of bases T, C, A, and G in the *COI* sequences were 25.3%, 18.1%, 28.1%, and 28.6%, and the content of A+T (53.4%) was higher than that of C+G (46.7%), showing an AT bias (Table 2).

### 3.2. Haplotype Distribution and Phylogeny

Basing on the *COI* sequence, the genetic diversity statistics showed that 11 haplotypes were detected in the 168 sampled *C. alburnus* individuals (Table 3 and Appendix Table A1). In detail, the TH population had the highest number of haplotypes, with 7; followed by the SH population, with 5; and the CJ and GH populations had the lowest number of haplotypes, with 2. Hap1 and Hap2 were the shared haplotypes in the five populations, and Hap2 appeared most frequently, distributed among 77 samples, accounting for 45.8% of the total fish individuals, and may be the original haplotype of *C. alburnus*. Among the unique haplotypes, there were three haplotypes in the SH population (Hap4, Hap5, and Hap6), one haplotype in the HH population (Hap3), and five haplotypes in the TH population (Hap7, Hap8, Hap9, Hap10, and Hap11). Based on Kimura’s two-parameter model, the ML phylogenetic tree of the 11 haplotypes was constructed with *C. dabryi* (GenBank accession number: EF467625.1) as the out group. From the evolutionary tree, the haplotypes were mainly divided into three branches. Hap6 and Hap11 as well as Hap4 and Hap9 each clustered into one branch, mainly from the SH and TH populations, and the remaining seven haplotypes clustered into one branch, including five populations (Figure 2).

Network 5.0 software was used to analyze the evolutionary relationships among the *C. alburnus* haplotypes (Figure 3). The haplotype network distribution of the five populations showed a radial structure centered on the shared haplotype, Hap2, of the five populations, with the largest proportion of Hap2 in the GH population (33.8%). The haplotypes of the other populations were linked by single mutations or multi-step mutations and showed a star-like distribution around the original haplotype, which was consistent with the results of the ML tree.

### 3.3. Genetic Diversity and Genetic Structure of the Population

The results of the genetic diversity parameters of the five populations of *C. alburnus* showed that the population haplotype diversity ranged from 0.280 to 0.746, and the nucleotide diversity index was 0.00053 to 0.01834 (Table 3). The TH population had the highest haplotype diversity and nucleotide diversity, while the GH population had the lowest haplotype diversity and nucleotide diversity. The results showed that the TH population had the highest genetic diversity, followed by the SH and CJ populations, and the HH and GH populations had the lowest genetic diversity.

The genetic distance analysis of the five populations showed that the genetic distance within the populations ranged from 0.00054 to 0.01992. The largest genetic distance within the populations was in the TH population, and the smallest genetic distance was in the GH population (Table 4). Based on the genetic distance between the five populations, an NJ tree was constructed, and the genetic distances between two populations ranged from 0.00073 to 0.01588, with the greatest genetic distance between the SH population and the TH population (Figure 4). The results of the inter-population Fst analysis showed that the Fst values ranged from 0.00035 to 0.51712, and the genetic variation between the SH population and other populations was greater, with the greatest genetic variation and significant differences (*p* < 0.05) with the CJ population and the smallest variation in the HH and GH populations (Table 4).

### 3.4. Analysis of Population History Dynamics of the Population

This study used a combination of neutral tests, a mismatch distribution, and a BSP analysis to analyze the historical population dynamics of *C. alburnus* from five different populations. Fu’s Fs detection value was 1.890; the mismatch distribution indicated that the *C. alburnus* population showed a multi-peaked pattern. It is speculated that the *C. alburnus* population showed a relatively stable state and that no population expansion occurred (Figure 5). To further estimate the historical population dynamics of *C. alburnus*, the overall BSP analysis was performed using BEAST software based on the average mutation rate of the animal’s mitochondrial *COI* gene, at about 1% per million years [22], and the results showed that the *C. alburnus* population showed a tendency to remain stable (Figure 6).

## 4. Discussion

### 4.1. Genetic Diversity of the Population

Genetic diversity is a prerequisite for species to adapt to environmental changes, and its level determines the long-term viability and evolutionary potential of species. The conservation of genetic diversity is of vital importance for the long-term survival of species. The genetic diversity of species is an important part of biodiversity research and reflects the ability of species to adapt and evolve in response to environmental changes. The higher the genetic diversity, the stronger the environmental adaptive ability [23]. Among the parameters that measure the genetic diversity of a species, nucleotide diversity is somewhat better than haplotype diversity [24]. According to the criterion proposed by Grant et al., the threshold value for haplotype diversity is 0.5 and the threshold value for nucleotide diversity is 0.005; the higher the value of both, the higher the degree of genetic diversity in the population [25]. In this study, the genetic diversity of 168 DNA sequences of five geographic populations of *C. alburnus* was analyzed, resulting in a total of 11 haplotypes, and the nucleotide diversity of the five populations ranged from 0.00053 to 0.01834, with the highest genetic diversity in the TH population, followed by the SH population, with lower genetic diversity in the HH, CJ, and GH populations. Compared to previous studies, the genetic diversity of the TH population was higher than that in the Pearl River (Pi = 0.007) [10], the Three Gorges Reservoir (Pi = 0.0141) [26], Xingkai Lake (Pi = 0.0041), and Danjiangkou (Pi = 0.0017) [27]. Genetic diversity is influenced by a variety of factors. Population evolution and historical factors are the main reasons for reduced levels of genetic diversity, and geographic isolation and inbreeding can lead to differentiation in gene exchange between populations, reduced gene richness, and severe homogenization. Overall, the *C. alburnus* population showed high haplotype diversity and high nucleotide diversity, and it is speculated that this situation may be due to the secondary contact of independent populations [28]. TH is the largest lake, with a high degree of protection and a prominent ecological status. *C. alburnus* has different ecological protections in the different lakes. The SH population is possibly affected by the active release of cultured fish, avoiding inbreeding and bottleneck effects and leading to population mixing and the random exchange of genes, resulting in increased genetic diversity. However, the CJ and HH populations have been overfished for many years, and the fishery resources have seriously declined. The habitats have changed, and the germplasm resources have declined. Under the influence of net-enclosure aquaculture facilities, GH has not yet fully restored its fishery resources, resulting in the low genetic diversity of *C. alburnus*.

### 4.2. Genetic Structure of the Population

Fst is an important indicator of the degree of genetic differentiation among populations [29]. In this study, the Fst values of five different *C. alburnus* populations were calculated to evaluate the genetic differentiation among populations, and the results showed that the Fst of the SH population was larger (Fst > 0.25) than those of the other four populations, with the highest degree of differentiation with the Changjiang River population (Fst = 0.51712). The genetic variation was mainly from within the population, and the intra-population variation was much greater than the inter-population genetic variation, which was consistent with the genetic differentiation coefficient (Fst) of the *C. alburnus* population, which showed negative values when the intra-population variation was greater than the inter-population variation [30].

Different geographic populations tend to have significant genetic differences due to isolation, and geographic isolation limits gene flow and leads to a high degree of genetic differentiation among populations [31]. From the NJ phylogenetic tree constructed using genetic distance, the five populations showed a significant geographical pattern. The NJ tree divided the five populations into three branches, with the HH, CJ, and GH populations in central China clustered into one and the SH population in the north and the TH population in the south distributed independently, with the SH and TH populations having the greatest genetic distance and maintaining consistency in geographic distribution. The large genetic difference between the SH population and the other four wild populations may be attributed to geographical distance. SH lake is very far from the other four regions. SH and TH are two different lakes that are independent of north and south China. Fish gene flow is mostly concentrated within their respective lakes or adjacent tributaries. Most fish are in relatively closed water bodies for a long time, and genetic exchange between the two populations is limited. Genetic differentiation occurred to form a distance isolation pattern, which is related to the lack of long-distance geographical distribution. The gene flow between populations, geographical isolation, and habitat heterogeneity are important factors in the formation of spatial genetic patterns in the population of *C. alburnus* [32].

### 4.3. Historical Dynamics of the Population

Historical population dynamics can reflect the influence of historical events such as past geological climate change and human activities on the current species distribution [33,34]. Generally, there is a belief that the negative and significant values of Tajima’s and Fu’s Fs in the neutral test indicate that the population expanded in history. The other result indicated that, based on the clear unimodal shape of the individual inter-basal mismatch distribution curve, population expansion can be considered [35]. The BSP analysis of the historical effective population size and the skyline plot of historical time for *C. alburnus* both indicated that the *C. alburnus* population was relatively stable and that the historical population dynamics did not significantly deviate from equilibrium. The neutral test showed that the overall Tajima’s D was negative and Fu’sFs test results were positive values. The mismatch analysis plot was a multi-peaked curve, which also indicated that the population was stable. The BSP further verified that the effective population of *C. alburnus* did not experience significant population expansion. In terms of historical evolution, the population experienced the last major ice age during the late Quaternary ice age 0.29 million years ago. Although alternating climate changes had a significant impact on the population, the Bayesian skyline diagram reflects a small effect on the effective population size of *C. alburnus* with the end of the last ice age and sea level rise due to climate warming. Populations of *C. alburnus* continued to spread outward, and in subsequent evolution and habitat changes different haplotypes were gradually generated, which formed a rich pattern of genetic structures.

## 5. Conclusions

In this study, a genetic diversity analysis of five different populations of *C. alburnus* was analyzed, and it indicated that populations with higher genetic diversitycan be managed by establishing conservation units. For the low genetic diversity of *C. alburnus,* populations should be enhanced through the breeding of *C. alburnus* germplasm resources and the protection of the survival environment. However, using individual mitochondrial sequence markers for population genetic analyses still has limitations, and subsequent studies should be conducted in combination with other methods such as the RAD (restriction-site-associated DNA) and GBS (genotyping by sequence) research methods to ensure more accurate results and an all-round assessment of the germplasm resource to provide data to support the sustainable and healthy development of *C. alburnus*.

## Figures and Tables

**Figure 1 biology-12-00012-f001:**
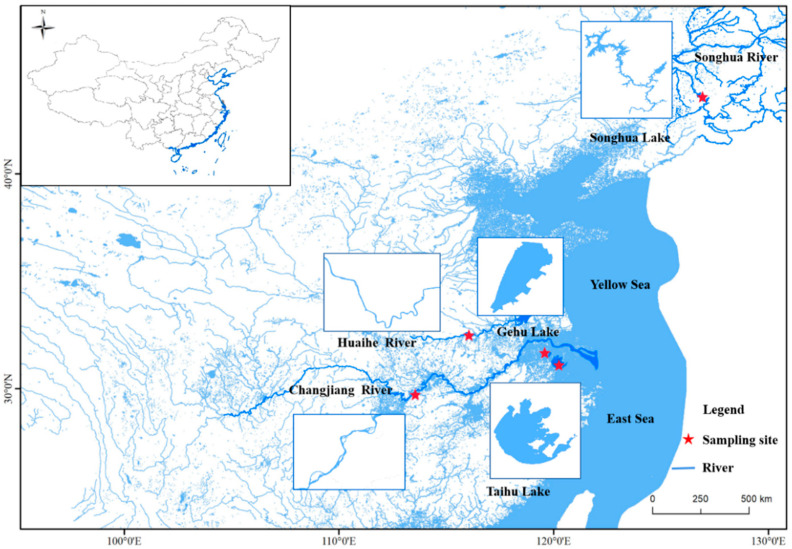
Sampling locations of *C. alburnus* in 5 different river basins.

**Figure 2 biology-12-00012-f002:**
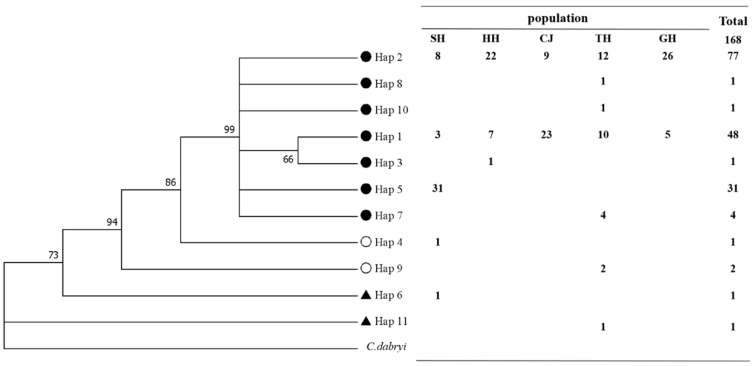
Distribution of 11 haplotypes in 5 populations and the ML tree. Note: the same symbol before haplotypes indicates clustering as a group.

**Figure 3 biology-12-00012-f003:**
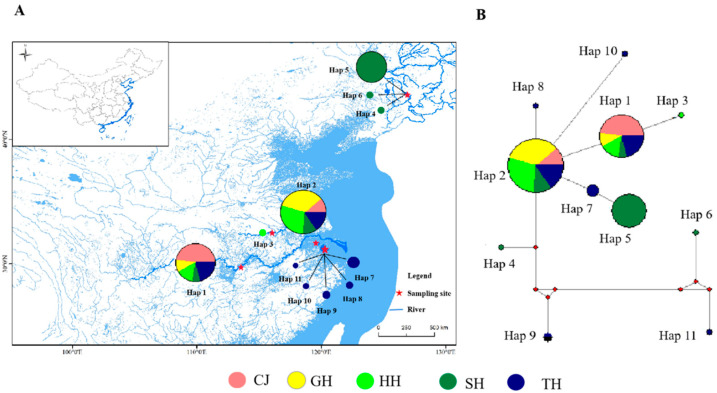
Geographical distribution of haplotypes (**A**) and haplotype network tree (**B**).

**Figure 4 biology-12-00012-f004:**
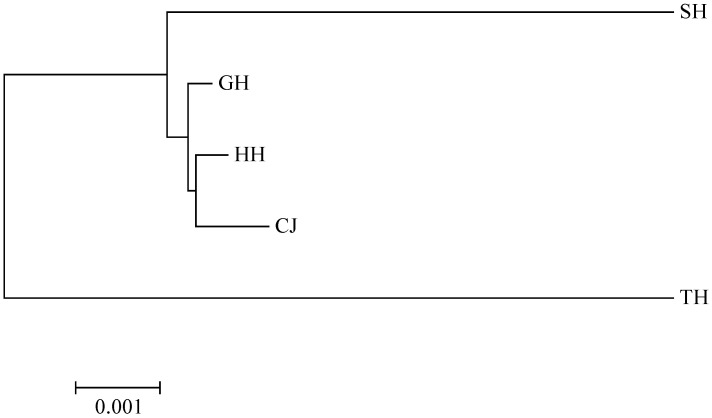
NJ trees of 5 populations based on genetic distance.

**Figure 5 biology-12-00012-f005:**
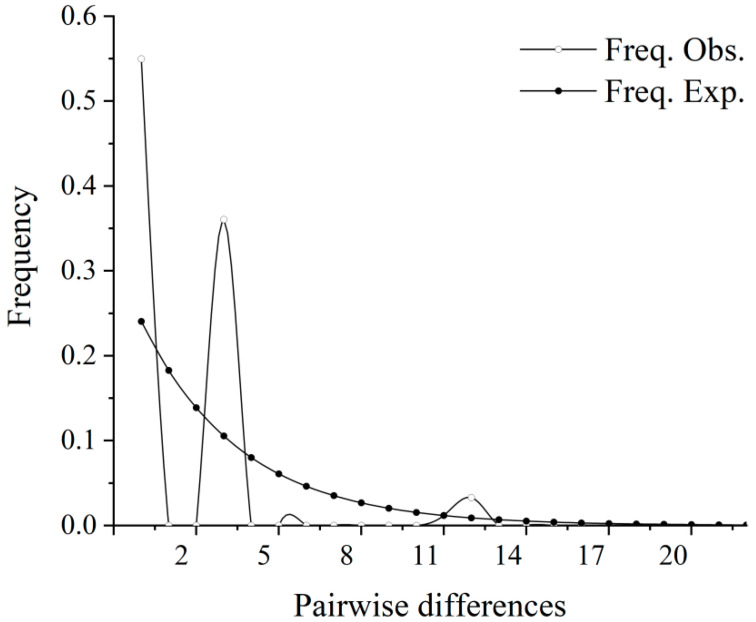
Mismatch distribution of *C. alburnus* based on *COI* gene sequences.

**Figure 6 biology-12-00012-f006:**
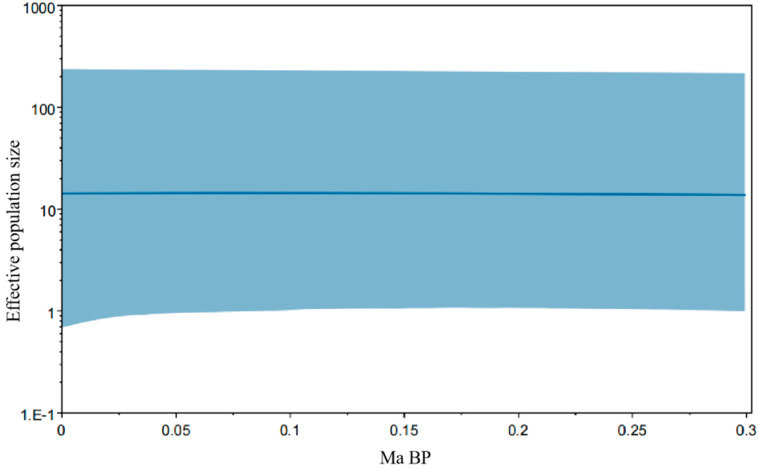
Bayesian skyline plot of *C. alburnus*.

**Table 1 biology-12-00012-t001:** Sampling information of *C. alburnus* in five different river basins.

Population	Abbreviation	Sample Size	Longitude	Latitude
Songhua Lake	SH	44	E 126°79′34.26′′	N 43°71′37.53′′
Huaihe River	HH	30	E 117°22′22.54′′	N 32°96′06.18′′
Changjiang River	CJ	32	E 116°27′72.95′′	N 29°79′57.50′′
Taihu Lake	TH	31	E 120°19′27.02′′	N 31°23′17.43′′
Gehu Lake	GH	31	E 119°82′67.59′′	N 31°66′33.95′′

**Table 2 biology-12-00012-t002:** Average base content of *COI* gene sequences of five populations of *C. alburnus*.

Population	Percentage of Nucleotide (%)					
	T(U)	C	A	G	T+A	C+G
SH	25.4	18.0	27.8	28.7	53.2	46.7
HH	25.2	18.1	28.1	28.5	53.3	46.6
CJ	25.1	18.2	28.1	28.5	53.2	46.7
TH	25.2	18.2	28.1	28.5	53.3	46.7
GH	25.3	18.1	28.1	28.5	53.4	46.6
Average	25.3	18.1	28.1	28.6	53.4	46.7

**Table 3 biology-12-00012-t003:** Genetic diversity of five populations of *C. alburnus* based on *COI* sequences.

Population	Haplotypes	Polymorphic Sites	Haplotype Diversity (*H_d_*)	Nucleotide Diversity (*P_i_*)	Neutrality Tests	
					Tajima’s D	Fu’s *Fs*
SH	5	64	0.476	0.00865	−2.52029	6.200
HH	3	2	0.421	0.00091	−0.21831	−0.143
CJ	2	1	0.417	0.00079	1.03928 *	1.328
TH	7	98	0.746 *	0.01834 *	−2.42247	7.380
GH	2	1	0.280	0.00053	0.18025	0.637
Total	11	115	0.677	0.00673	−2.69413	1.890

* indicates significant difference (*p* < 0.05).

**Table 4 biology-12-00012-t004:** Genetic distance and Fst of five different populations.

Population	SH	HH	CJ	TH	GH
SH	**0.00849**	0.47062 *	0.51712 *	0.29755 *	0.49043 *
HH	0.00669	**0.00092**	0.22271	0.02528	0.00035
CJ	0.00740	0.00124	**0.00081**	0.05308	0.46155 *
TH	0.01588 *	0.01068 *	0.01087 *	**0.01992**	0.03503
GH	0.00644	0.00073	0.00126	0.01056 *	**0.00054**

The value in bold in a diagonal line indicates the genetic distance within a population. The genetic distance between populations is below the diagonal line, and the Fst between populations is above the diagonal line. The “*” indicates a significant difference (*p* < 0.05).

## Data Availability

Not applicable.

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
