# Peer review of "Genetic Diversity Evaluation and Conservation of Topmouth Culter (Culter alburnus) Germplasm in Five River Basins in China"

_biology, 2022, doi:10.3390/biology12010012_

Round 1
Reviewer 1 Report
In the manuscript, “Genetic Diversity Evaluation and Conservation of Topmouth culter (Culter alburnus) Germplasm in five rivers Basins, China”. The authors used the COI gene as a molecular marker to depict the genetic characteristics, population structure, and population differentiation among five populations of Culter alburnus in north and south China. The results were significant for the conservation of Culter albunus germplasm resources. However, there were some drawbacks in the manuscript that needs to be fixed.
Major comments
1. In this manuscript, the authors tried to unveil the population structure, and genetic differentiation among five selected populations of Culter alburnus by using the COI gene, however, from my point of view, I think only the COI gene is not sufficient for the analysis. Although COI is a good marker for population genetics, there was still some bias. I would suggest more nuclear markers.
2. For Table 4, there were 98 polymorphic sites of the COI gene in the Taihu lake population, there were nearly 20% of sequence variations (96/526bp), however, only 1 site in Changjiang and 1 in Gehu lake. Taihu and Gehu are very close to each other and they both belong to the Changjiang river region. Is it reasonable? Could you give some discussion on this? Strong selection pressure on Taihu?
3. There were some inconsistent results, the Fst between Changjiang and Gehu is 0.46155 (very high!), and between Taihu and Gehu is 0.03503, which indicates that Taihu and Gehua have low genetic differentiation, however, for figure 3, Changjiang is more close to Gehu, which is very odd.
Minor comments:
1. In “Simple Summary” part, what do you mean by “species differentiation”?
2. In 2.3 section, what do you mean by “mean nucleotide (K)”?
3. You mentioned “strong AT bias” in the COI gene, however, I could not find a large difference in the AT and GC content (53.4 versus 46.7) in Table 2.
4. In table 3 which I think should be figure 2, what does the green circle, and blue triangle mean?
5. In table A1, what is the number mean? Location of variable sites or number of variable sites?
Reviewer 2 Report
In this article, the authors revealed the genetic diversity of Culter alburnus from five river Basins in China. However, unfortunately, I have many criticisms about the structure.
Introduction
- Introduction and research strategies don't provide enough background and should be explained in simpler and clearer language. But I think if recast could be improved. So, the introduction must be completely rewritten anticipating the object of the study and framing it into the context of previous results in a better manner, also from an experimental point of view.
Material and Methods
- The descriptions of the methods must be simplified as they are too long unnecessarily. Especially the explanation of genetic diversity analysis methods is not satisfactory and should be rewritten.
Results
- Likewise, the results were not presented clearly and unequivocally as they were needlessly prolonged, and very complex information was included.
- Table 5 should be rearranged. Fst values between populations above the diagonal line should be removed and statistically significant differences between populations should be indicated with (*) at values below the diagonal.
- Intra-population genetic distance values should be placed on the diagonal as bold.
- Besides, all the genetic distance and diversity values in Tables 4 and 5 should be arranged in a uniform way to be either 4-digit or 5-digit and corrected accordingly in the text.
- Table 6 should be removed. The percentage of variations is sufficient to explain in a sentence, there is no need for a table.
- Instead of UPGMA Tree, which is a simple, fast but unreliable method that generally produces equal branch tips assuming equal evolution rates, Neighbour Joining Tree should be generated, which allows unequal evolution rates, whose branch lengths are proportional to the amount of change, which is relatively faster and gives better results compared to the UPGMA.
- This change should be mentioned in 2.3 Genetic diversity analysis section.
Discussion
- Discussion part is very poor. None of the data obtained has been adequately discussed. It sounds more like a literature review regarding the topic than a real discussion of the results obtained. A comparison of the results collected in this work to those collected by other authors should be done in a more critical manner, commenting on similarities and accounting for differences. However, the manuscript in its present form does not report original experimental work or robust results.
Conclusion
- Conclusion is supported by the findings but should be simplified to a single paragraph instead of 2 paragraphs.
Apart from the comments about the body of the text mentioned above, the article unfortunately is not well written, and I detected many major grammatical and semantic errors. So, I strongly suggest that the article must undergo an extensive English editing process by an expert in this field.
Consequently, this study is suitable for a population genetics study in terms of the research design, genetic analysis, and bioinformatic interpretations. However, I think that the article may be suitable for publication in its final version, provided that almost all of it has been simply and clearly rewritten and sufficiently revised by the authors, and finally a comprehensive English editing process.
Reviewer 3 Report
In the results section, there is confusion between Table #5 and the text. Textual result is not exactly reflect the Table numbers (also highlighted in Text ) .

Round 2
Reviewer 1 Report
no new comments on the revised manuscript
Reviewer 2 Report
I thank the authors for kindly considering my suggestions and reshaping the article accordingly. Finally, there are a few issues that need to be corrected.
-Authors should ensure that the species mentioned as the outgroup with GenBank accession number belongs to C. dabryi shinkainensis. Because this accession number (MF805651.1) given in the article belongs to Culter alburnus in GenBank. Therefore, this complexity needs to be compensated. Authors must either specify a new outgroup species or provide the correct accession number. After the correction is made, the species name in Figure 2 should be corrected accordingly.

Round 3
Reviewer 2 Report
The final version of the M&S is well designed by the authors and can be published. Thanks to the authors for correcting in line with my suggestions.